# A Multi-Agent Approach to Binary Classification Using Swarm Intelligence

Sean Grimes * and David E. Breen 

Department of Computer Science, Drexel University, Philadelphia, PA 19104, USA
* Correspondence: spg63@drexel.edu

**Abstract:** Wisdom-of-Crowds-Bots (WoC-Bots) are simple, modular agents working together in a multi-agent environment to collectively make binary predictions. The agents represent a knowledge-diverse crowd, with each agent trained on a subset of available information. A honey-bee-derived swarm aggregation mechanism is used to elicit a collective prediction with an associated confidence value from the agents. Due to their multi-agent design, WoC-Bots can be distributed across multiple hardware nodes, include new features without re-training existing agents, and the aggregation mechanism can be used to incorporate predictions from other sources, thus improving overall predictive accuracy of the system. In addition to these advantages, we demonstrate that WoC-Bots are competitive with other top classification methods on three datasets and apply our system to a real-world sports betting problem, producing a consistent return on investment from 1 January 2021 through 15 November 2022 on most major sports.

**Keywords:** prediction; swarm; multi-agent; Wisdom-of-Crowds; sports-betting; collective-intelligence

## 1. Introduction

We present extensions to a multi-agent-based system for binary classification, capable of being distributed across multiple compute nodes, using agents each with partial knowledge of the information space. The system is based on Wisdom of Crowds (WoC), and uses a honey-bee-derived swarm mechanism for opinion aggregation, that provides a prediction and associated confidence value. The multi-agent design allows new features to be incorporated into classification problems without re-training the existing agents, reducing the computational costs and training time of adding new feature to an existing model.

Artificial neural networks (ANNs), specifically deep neural networks (DNNs), and ensemble learning methods are the current 'state-of-the-art' approaches to classification problems [1]. Both families of methods, however, require a computationally expensive re-training operation to incorporate additional features, i.e., if the number of inputs to the network changes [2]. Some specialized types of convolutional (CNN) and recurrent (RNN) neural networks allow for variable length input for image and time-series data (respectively), but add significant complexity over typical DNNs and do not generalize to binary classification problems well [3,4]. Transfer learning allows a classification model developed for one task to be used for a different, but related, task and can incorporate an additional intermediate input layer to process additional input features [5]. Transfer learning adds complexity to a network and does not guarantee that an existing model will perform well on the new problem [6]; the features used to train the original model may not be predictive for the new dataset [7].

A multi-agent-based classification system has been previously reported in Grimes et al. [8], and is based on prediction markets (PMs), WoC, and swarm intelligence—the collective intelligence or behavior of simple, decentralized agents. PMs are built to determine the probability of a future event taking place by collecting truthful input from participants, aggregating the input and forming a collective knowledge [9]. In order to accomplish this,

PMs expect all participants to be well-informed agents, a very difficult requirement to implement with computer-based agents in any non-trivial system. Expert computer-based agents are difficult to develop, both from a programming and information availability standpoint; how can you program an 'expert' agent without having significant subject matter knowledge yourself? Othman said on computer-based agents "agent-based modeling of the real world is necessarily dubious. Attempting to model the rich tapestry of human behavior within economic structures—both the outstandingly bad and the terrifically complex—is a futile task" [10].

Wisdom of Crowds is an alternative prediction method which performs better with an information-diverse pool of participants and does not require expert participation. Predictive error decreases as information diversity increases [11]. WoC achieves this behavior by requiring a competent opinion aggregation mechanism that takes input from all participating agents and outputs an overall prediction. Previous work in this area has investigated the Unweighted Mean Model (UWM), Weighted Voter Model (WVM) and other aggregation mechanisms [12–14]. The classification method used in our work employs a more complex, honey-bee-derived swarm optimization algorithm as the WoC aggregation mechanism, which has previously shown to improve classification performance compared with other methods and allows for a confidence score to be associated with each prediction [8].

In this paper, we show how this method compares with other common classification methods using three datasets, predicting the metastasis status of breast cancer patients, the success of Hollywood movies, and the satisfaction of airline passengers. We demonstrate that additional features can be included without re-training existing agents, which improves overall classification performance. Additionally, we show this algorithm can be distributed across multiple compute nodes, and describe the challenges with distributed opinion aggregation and our approach to solving those challenges. We also include real-world results when applying this method to sports betting across multiple sports and bet types.

Section 2 describes the overall system design, how additional features are added to the classification problem, a 'meta-swarm' method, and how the system is distributed across hardware nodes. Section 2 also describes the datasets used in testing and how our system was applied to a sports betting problem. Section 3 presents comparisons with our WoC-Bots and 'meta-swarm' methods with results from AdaBoost, XGBoost, Random Forest, a Deep Neural Network, and Logistic Regression for three datasets. Also presented are results for including additional features, distributing our agents across hardware nodes, and multi-year testing on sports betting. Section 4 discusses the presented results and reviews the benefits to a multi-agent design. Section 5 reiterates our approach using WoC-Bots and swarm intelligence in addition to presenting possible ideas for future work.

## 2. Materials and Methods

### 2.1. System Design

The classification method uses simple agents (WoC-Bots) without expert knowledge. At the core of each agent is a simple multi-layer perceptron (MLP) classifier with 1 to 4 hidden layers—depending on input size—generally rounded to the nearest integer value of $input\_size * 0.3$. The goal is for the MLP to be as shallow as possible while still extracting features from the input. For the datasets we have tested, using the formula above has proven successful given each agent only receives a small subset of the available input. The rest of the agent is designed as a modular system for efficiently modifying agent behavior. Each agent's MLP classifier is trained with a different, small, subset of features. This initially gives the group of agents a diverse set of knowledge. All agents with relevant knowledge for the classification task then interact with one another, share knowledge, determine the trust they have in other agents and the confidence in their own opinion, and change their opinion given enough evidence.

As with work described in [8,14], we use the DeepLearning4j (DL4J) [15] implementations of the Adam updater [16], softmax activation function [17], and standard stochastic gradient descent for the MLP implementation and deep neural network. DL4J is "an

open-source, distributed deep-learning project in Java and Scala spearheaded by the people at Skymind, a San Francisco-based business intelligence and enterprise software firm". All code was written in Kotlin (versions 1.3.20–1.3.41), running on the Java Virtual Machine (version 11). All features, agent history data, and trained agents were stored in SQLite3 databases. IntelliJ IDE was used for our development environment.

Prior to the interaction and opinion aggregation steps, each agent receives 1 to 4 highly correlated features in common between all agents. Feature correlation was determined using principal component analysis, run externally to the system being described here. Other available features were distributed across multiple agents randomly, with duplicate features and multiple agents with the same feature set neither limited nor encouraged. Agents that show poor MLP classification performance during the initial training step are not included in the remainder of the classification steps. Poor performance was defined as less than 50% accuracy on the test set.

### 2.1.1. Interaction Period

Table 1 presents the attributes which describe the internal state of an agent during the interaction period. Many of these values are available to other agents during interactions within the social interaction arena. The `confidence` attribute can be biased to favor accuracy, precision, or recall metrics depending on which metric is the most important for the given classification problem. For example, a classification task such as the breast cancer dataset we present in the results section may benefit from biasing towards recall to better avoid false negative predictions

$$confidence = accuracy * 0.25 + precision * 0.25 + recall * 0.50. \tag{1}$$

**Table 1.** Internal scoring variables.

| Variable | Description |
|---|---|
| `current_prediction` | Binary, class 0 or class 1. |
| `trust_score` | Updated during interaction based on agreement and performance. |
| `features` | A list of features used by the agent's classifier. |
| `prior_performance` | Long-term history of agent performance, varied between 0.7 and 1.3 where 1.0 is average performance. |
| `certainty` | Initialized to MLP classifier performance, updated during interaction period. Represents how strongly the agent believes in the current prediction. |
| `eval_accuracy` | Initial classification accuracy. |
| `eval_precision` | Initial classification precision. |
| `eval_recall` | Initial classification recall. |
| `confidence` | Biased value based on accuracy, precision, and recall. |

The interaction arena can take any 2D shape comprised of square faces. In this work, the arena is a square with the number of available discrete spaces available to the agents equal to 2 times the number of participating agents. Agents are initialized randomly inside the arena and agents are not allowed to share spaces with other agents at initialization. After initialization, up to two agents can share the same space, which starts an interaction between those two agents. Following the interaction, agents are moved within the arena in a "Manhattan-like" fashion, moving either one space north, south, east, or west within the bounds of the arena. The direction is randomly selected and agents cannot interact with each other two times in a row or more than twice within five separate movements. Agents who cannot move because of these restrictions are 'teleported' to a randomly selected

empty space within the arena. Agents are also 'teleported' to a random empty space every 10–15 iterations to further facilitate information dispersal.

The interaction period is operated in discrete iterations; each agent is moved during each iteration and all interactions between agents must complete before the next iteration can begin. The number of iterations is based on the number of participating agents where

$$interaction_{iters} = num\_agents * 0.10 \tag{2}$$

to balance run time with information dispersal. Any agent with missing data for the current classification task does not participate; all agents in the interaction arena and subsequent steps have complete information for the features they were assigned for the current classification question.

During the interaction period, two agents, agent *a* and agent *b*, interact when sharing a space within the arena. The following equations govern the updates to the internal state of agents during the interaction and determine if and when an agent's prediction will change, and what confidence the agent has in its current prediction. The equations assume agent *a* is receiving information from agent *b*, however during an interaction the inverse will also occur.

Once two agents meet, agent *a* first determines how willing it is to accept information from agent *b*, a function of *a*'s current `certainty`, where $a_{certainty}$ represents *a*'s current `certainty` and $a_{acceptance}$ represents *a*'s willingness to accept information from agent *b*

$$a_{acceptance} = 1.0 - a_{certainty}. \tag{3}$$

Agent *a* determines how much influence to allow agent *b*, $b_{influence}$, a function of agent *b*'s `confidence`, `trust_score`, and `certainty`, and agent *a*'s acceptance, $a_{acceptance}$,

$$b_{influence} = b_{confidence} * a_{acceptance} * b_{trustCertainty}, \tag{4}$$

where $b_{trustCertainty}$ represents `trust_score * certainty`,

$$b_{trustCertainty} = b_{trustScore} * b_{certainty}. \tag{5}$$

Agent *b*'s influence is further modified based on agent *b*'s `prior_performance`, represented by $b_{priorPerf}$, such that the corrected influence can be represented by $b_{correctedInfluence}$,

$$b_{correctedInfluence} = b_{priorPerf} * b_{influence}. \tag{6}$$

If agent *a* and agent *b* have different predictions, $b_{correctInfluence}$ will be multiplied by $-1$ to act against agent *a*'s current predictive belief. Agent *a*'s updated certainty, $a_{certainty}$, can be calculated by Equation (7), where *a*'s certainty increases if both agents *a* and *b* have the same prediction and decreases if the predictions are different,

$$a_{certainty} = a_{certainty} + b_{correctedInfluence}. \tag{7}$$

If agent *a*'s certainty value falls below 0.50, agent *a* will change its prediction. If the prediction is changed, agent *a*'s certainty is updated by Equation (8),

$$a_{certainty} = 1.0 - a_{certainty}. \tag{8}$$

### 2.1.2. Swarm Aggregation

Opinion aggregation in WoC systems is an open problem [18]. Previous work has shown promise in aggregating human opinions [19,20], and the work presented here continues with the honey-bee-derived "swarm intelligence" aggregation mechanism published in [8]. Bees and other simple organisms are able to forage large areas, finding near optimal food sources and new nesting locations by sending a subset of bees, the scouts, to search the environment around the colony [21]. Returning scout bees will advertise the location

of their find using a "waggle dance", with the movements indicating direction relative to the sun and the length of dance thought to be proportional to the quality of the discovery. Scouts which dance for longer periods of time generally recruit more bees to their discovery [22]. As long as a discovered food source remains high quality, the bees will continue to advertise it, allowing it to remain productive. Advertisement of the food source will naturally decline as it becomes depleted and fewer scouts advertise the location [23].

The computer-based version of the honey-bee foraging algorithm operates similarly. The swarm aggregation step is designed for all agents to support a single binary opinion, either 1 or 0. All of the agents which participated in the social interaction step must also participate in the swarming step. Initially, 20% of agents are randomly selected to be "presenters", similar to the "scout" bees in nature. The remaining agents are assigned as "watchers", which are assigned to the presenting agents. Initial assignment of "watchers" is accomplished using 'fitness proportionate selection' [24]. Each of the presenting agents has an assigned probability, $a_{prop}$, between 0 and 1, where the sum of all probabilities of all presenting agents is 1 after normalization. The probability for each agent is calculated by Equation (9), an equally weighted combination of the presenting agent's `prior_performance`, $a_{priorPerf}$, `confidence`, $a_{confidence}$, and `trust_score`, $a_{trust}$.

$$a_{prob} = (a_{priorPerf} + a_{confidence} + a_{trust})/3 \qquad (9)$$

The 'fitness proportionate selection' algorithm uses the computed probabilities for each of the presenting agents to assign "watchers" to be supporters of each of the presenting agents using $a_{prob}$, such that presenting agents with a higher $a_{prob}$ will more frequently be assigned "watchers" than those with a lower $a_{prob}$ value. "Watchers" can only be assigned to a single presenting agent at any given time and 'support' that presenting agent simply by being assigned to it. However, each "watcher" can request re-assignment two times if its prediction is different from that of the presenting agent it was assigned and if its $a_{priorPerf}$ value is higher than the presenting agent's, allowing a high performing agent the opportunity to move to an agent it thinks better represents its prediction. However, this process uses the same fitness proportionate selection algorithm, which offers no guarantee that a move will be seen as beneficial to the watching agent. The limit of two moves accomplishes three things, reducing compute time, preventing a potential infinite loop of constant movement, and introducing further diversity of opinion where agents may be assigned to support an agent they disagree with. Once assignment and agent movement is complete the presenting agent represents itself and the watchers it was assigned; e.g., if a presenting agent is assigned 10 watchers that presenting agent's opinion is now worth 11 'votes' during the final aggregation process.

The "presenters" and "watchers" now represent a swarm of agents which goes through iterative steps to arrive at a collective binary prediction with an associated confidence value by repeating the process of assigning watchers to presenters and taking a vote of all presenting agents. While this process is running, the decision threshold is iteratively lowered if a higher threshold has not been met. The initial threshold is 100% agreement directly following the initial process of assigning watchers to presenters and taking a vote of all presenting agents. If this threshold is met, the prediction is considered "Very High Confidence", the prediction and confidence are returned and the swarming period ends. When there is no immediate agreement, agents perform an interaction period with all of the "watchers" assigned to the "presenter" and with the "presenting" agent. This interaction step follows the same protocols outlined in Section 2.1.1. This step facilitates additional information dispersal between agents. Following the interaction step, a new group of presenting agents is randomly selected and the agents go through the same steps as previously outlined. This process can run for an additional 100 iterations with the threshold for agreement lowered to 90%; if the 90% threshold is met at any point the prediction is considered "High Confidence" and the swarming period ends.

When neither of the above thresholds are met, the decision threshold is further reduced to 75% for an additional 50 iterations of selection, voting, and interaction. If the

75% threshold is met the prediction is "Medium Confidence". However, if the threshold has still not been met, a weighted vote is taken from all presenters and watchers, and the prediction is "Low Confidence". The weighting is based on the agent's `certainty` score, with agents more certain in their opinion given more weight than those who are less certain. The confidence thresholds were initially selected using the breast cancer dataset described in Section 2.2.1 based on experimentation with the goal of 100% accuracy for the "Very High Confidence" interval. The number of iterations for each threshold was selected to balance classification performance with runtime. A future goal is to automate the selection of threshold values for each dataset and classification constraints. For example, some usage scenarios may prefer more samples captured in the "Very High Confidence" interval while having a lower average accuracy for that category.

The confidence categories can be summarized as:

- Very High Confidence: All presenting agents agree on a prediction class immediately following fitness proportionate selection,
- High Confidence: 90% of presenting agents agree on a prediction class within 100 iterations of the swarming process,
- Medium Confidence: 75% of presenting agents agree on a prediction class after 100–150 additional iterations of the swarming process,
- Low Confidence: Weighted vote of presenters and watchers if above thresholds are not met.

### 2.1.3. Additional Features

Another benefit of the multi-agent approach to classification versus a traditional monolithic neural network is the ease at which new input features can be added to the computation. While some convolutional networks can accept variable length inputs, most traditional networks can only be expanded vertically with new layers while holding existing weights stable. They cannot expand horizontally, i.e., to include previously unknown input feature(s), at the input layer without re-training the network [25]. While each of our agents has a fixed number of inputs at creation, an internal MLP network restriction, the interactive and swarm steps of the overall system do not have restrictions on the number of participating agents. As noted in Section 2.1.1, agents with missing data simply do not participate. A natural extension to the variable number of participating agents is the ability to include newly generated agents representing the new data, a feature that does not require the re-training of the existing agents.

When new features become available, we generate new agents using a combination of the new features and the existing features. Combining the new and existing features in the newly generated agents solves two problems; (1) new agents need to know about the features that the existing agents are trained on, and (2), the existing agents need a way to determine `trust_score` and `prior_performance` metrics for the newly generated agents. Both the social interaction arena and swarm components of the system rely on social variables built around agent prior performance, trust, and existing knowledge of features. Newly generated agents derive estimated `prior_performance` and `trust_scores` values from the existing features they receive. The estimates come from averaging these values from existing agents that have similar features. The averaging step does have some shortcomings, particularly when agents with similar, but not identical, features are used to generate the estimates; the estimated values may not accurately reflect the correct `trust_score` and `prior_performance` of the newly generated agents. However, the new agents have shorter interaction and prior performance histories compared with existing agents, which allows for larger initial modifications to these metrics during the interaction step, typically correcting any poor estimates within a few interaction periods.

Incremental features addition was tested using two datasets, one attempting to predict breast cancer metastasis status by including additional measurements of cellular morphology data derived from whole-slide images of a representative hematoxylin and eosin (H&E) stained slide of the primary tumor. These data are in addition to the primary dataset as

described in Section 2.2.1. The second dataset that was used for testing attempted to predict the success of Hollywood movies. The initial predictions were made with a combined dataset from The Movie Database (TMDb) and MovieLens using only data available in both datasets. Additional features were included from MovieLens, which included cast information beyond the leading cast member, as well as information about the production company and producers. More information about these data are available in Section 2.2.2.

### 2.1.4. Meta-Swarm

An additional use of the multi-agent design and aggregation mechanism was developed to incorporate predictions from other sources. Sources can be other classification methods or direct predictions. This is conceptually similar to stacking in ensemble learning, where multiple weak learners predictions' are used as input to a 'meta-learner' and a regression method is used to combined the predictions from the weak learners [26]. However, in the system presented here, the original prediction from WoC-Bots and predictions from other sources are encapsulated into agents, replacing the MLP step as originally described in Section 2.1. As previously described, the agents are initialized in the interaction arena, interact socially to share information and develop trust, and an aggregate prediction with a confidence value is made following the swarm step.

This method has been applied to the breast cancer, Hollywood success, and airline satisfaction datasets using input from WoC-Bots, XGBoost, AdaBoost, RandomForest, Logistic Regression, and a Deep Neural Network. Our method was also used in a real-world application of the system for sports betting, using WoC-Bots and independent prediction sources as input. Previous testing has indicated that additional information sharing during the interaction phase improves overall predictive accuracy, precision, and recall [14]. Therefore, while applying this method, each input prediction was represented by five identical agents to facilitate additional information sharing within the interaction arena. Testing indicated no additional predictive benefit to representing each prediction with more than five agents. Agents which represented the same input prediction were allowed to interact with each other. Detailed results using the meta-swarm approach can be seen in Section 3 for the three datasets tested and the sports betting application.

### 2.1.5. System Distributability

The multi-agent nature of this system lends itself well to being distributed across multiple computational nodes using a cluster of otherwise independent machines. There are three main phases of this work to consider for distributability, the per-agent training phase where each small MLP network is trained, the social interaction phase, and the swarming phase where agents interact and vote before a system-wide prediction is made.

The training phase relies on a small amount of data which can be made available on a node-by-node basis or transferred across the network without significant delay. During this phase each agent is independent and distribution is straightforward. Agents can be placed on each node randomly, or using an initialization algorithm to optimally pick a node for each agent based on computational load or network limitations. The MLP network for each agent can be trained independently from any other agent and the small size and depth of the network do not limit training to powerful hardware.

The social interaction phase initializes all participating agents in the interaction arena. The flexible design of the arena allows each compute node to represent some subdivision of the full arena, e.g., an arena with 4 nodes and 120 possible spaces will be subdivided so that each node is responsible for a block of 30 spaces. An agent moving between each subdivision is moved between nodes. The movement algorithm governing each agent can encourage or discourage movement between nodes to balance information dispersal with network transit times. Figure 1 shows a possible configuration with a single interaction arena split across 16 nodes. The dark outline represents the bounds of the arena and the lighter internal lines represent boundaries between nodes. Agents can move between nodes

freely during the interaction period, which requires network transfers, either explicitly from some node to another node, or implicitly through shared storage.

The work presented here takes advantage of a disk-based database and a networked Ceph storage cluster that is shared by each node to implicitly share data on each node without explicit network transfer; no agent is serialized for transfer across the network stack. After each iteration in the interaction arena, the state of each agent is flushed to an SQLite database, which is updated and accessible to all nodes using the networked Ceph storage cluster. As each agent crosses a boundary between nodes, a per-agent flag is set indicating on which node the agent is present. Each node checks a table to determine which agents are present on that node prior to the start of every interaction iteration.

| N0 | N1 | N2 | N3 |
|---|---|---|---|
| N4 | N5 | N6 | N7 |
| N8 | N9 | N10 | N11 |
| N12 | N13 | N14 | N15 |

**Figure 1.** Representation of the full interaction arena split across 16 nodes.

The swarming phase of the system can require frequent movement of agents between nodes due to the frequent re-selection process for "presenters" and "watchers" while the swarm is iterating. The selection process randomly selects "presenters" and then distributes agents to available nodes. This can reduce the benefit of distributing this method across multiple compute nodes, due to the additional run time cost from network transfers.

The swarming phase of the system can work in two distinct modes of operation. The first option requires a simple command and control node, which is responsible for selecting the "presenters" and "watchers", as described in Section 2.1.2, and uniformly distributes the agents to available nodes. Each additional iteration of the swarming process does, however, require global knowledge and control to re-distribute agents to each node. This method does produce results identical to swarming on a single node, but is the slower of the two options.

The second option allows localized swarming on each node with aggregation at the end of the swarming process. Following the interaction phase, the agents can either stay on their current node, or agents can be randomly distributed to nodes. There is no additional movement of agents between nodes throughout the swarming process. In order to accomplish this, a method of representing each node's prediction was developed. Initial methods attempted to assigned a weight to each node based on multiple factors, such as confidence category for the node's prediction, the average prior performance of agents on the node, the average prior performance of the top 15% of agents on the node, and the

average confidence of all agents on the node and the top 15% of agents on the node. Of these methods, assigning a node weight based on average performance of the top 15% of agents produced the best results, however the accuracy of this method did not match the the previous option or single-node swarming.

Another method to represent each node was implemented and tested based on the meta-swarm that was described in Section 2.1.4. As with the above option, each node receives a random distribution of agents and the swarming process happens locally on each node. When the swarming process is complete each node's prediction is encapsulated into an agent and these agents are moved to a single node to interact and swarm before a final prediction is made. While this does require a single node capable of running the swarming process, the maximum number of agents is equal to the number of nodes and each agent is computationally simple, this should not present computational limitations for any node that is also capable of participating in the swarming process. This method has shown similar performance to the first, network-intensive option, on two of the three datasets presented in this paper.

### 2.2. Datasets

The following breast cancer metastasis status, Hollywood movie success, and airline satisfaction datasets were used to test our method against existing 'state-of-the-art' classification methods, including XGBoost [27], AdaBoost [28], RandomForest [29], Logistic Regression [30], and a DL4J DNN implementation. These methods were selected for comparison based on their state-of-the-art status and common use in binary classification tasks [1]. Logistic Regression was also considered for comparison to show performance against a pure statistical probability model. These datasets were also used to demonstrate the incremental feature addition component of our agent-based system. The sports betting data were used to demonstrate the flexibility of the agent-based design, making an initial classification prediction using data as described in Section 2.2.4, then including that prediction as a series of agents in a meta-swarm that utilizes predictions from other sources, aggregates the agent opinions, and renders a final prediction.

### 2.2.1. Breast Cancer Metastasis Status

We applied our agent-based method to predict lymph node metastasis status, node-positive or node-negative, in 483 de-identified breast cancer patients using all features which were available for at least 50% of patients. Previous work in this area considered only the most highly correlated features and required feature completeness [31,32]. The `confidence` attribute for all agents was biased to prefer high recall scores to better avoid false negatives, specifically $(0.25 * accuracy, 0.25 * precision, 0.50 * recall)$. Table 2 describes the clinical features available for use in the classification task. The additional cellular morphology features available for classification are:

- Cellular mean area
- Cellular mean circularity
- Cellular mean eccentricity
- Cellular mean intensity
- Standard area
- Standard circularity
- Standard eccentricity
- Standard intensity

**Table 2.** Characteristics of patient populations.

| Feature | $n = 483$ | % |
|---|---|---|
| Age | | |
| ≤45 | 103 | 21.3 |
| >45 | 380 | 78.7 |
| PT Max size (mm) | | |
| ≥200 | 5 | 1.0 |
| 100–199 | 15 | 3.1 |
| 50–99 | 51 | 10.6 |
| 25–49 | 125 | 25.9 |
| 0–24 | 271 | 56.1 |
| unknown | 16 | 3.3 |
| Angio Lymphatic Invasion | | |
| Absent | 127 | 26.3 |
| Present | 200 | 41.4 |
| Unknown | 156 | 32.3 |
| pT Stage | | |
| Unknown | 36 | 7.5 |
| pT1 | 210 | 43.5 |
| pT2 | 173 | 35.9 |
| pT3/pT4 | 64 | 13.3 |
| Histologic Grade | | |
| Unknown | 33 | 6.8 |
| 1 | 53 | 11.0 |
| 2 | 164 | 34.0 |
| 3 | 233 | 48.2 |
| Tubule Formation | | |
| Unknown | 30 | 6.2 |
| 1 (>75%) | 13 | 2.7 |
| 2 (10–75%) | 98 | 20.3 |
| 3 (<10%) | 342 | 70.8 |
| Nuclear Grade | | |
| Unknown | 29 | 6.0 |
| 1 | 20 | 4.1 |
| 2 | 151 | 31.3 |
| 3 | 283 | 58.6 |
| Lobular Extension | | |
| Unknown | 202 | 41.8 |
| Absent | 147 | 30.4 |
| Present | 134 | 27.7 |
| Pagetoid Spread | | |
| Unknown | 213 | 44.1 |
| Absent | 177 | 36.6 |
| Present | 93 | 19.3 |
| Perineureal Invasion | | |
| Unknown | 267 | 55.3 |
| Absent | 186 | 38.5 |
| Present | 30 | 6.2 |
| Calcifications | | |
| Unknown | 115 | 23.8 |
| Absent | 126 | 26.1 |
| Present | 176 | 36.4 |
| Present w/ DCIS | 66 | 13.7 |
| ER Status | | |
| Unknown | 51 | 10.6 |
| Negative | 155 | 32.1 |
| Positive (>10%) | 277 | 57.3 |

**Table 2.** *Cont.*

| Feature | $n = 483$ | % |
|---|---|---|
| PR Status | | |
| Unknown | 54 | 11.2 |
| Negative | 201 | 41.6 |
| Positive (>10%) | 228 | 47.2 |
| P53 Status | | |
| Unknown | 81 | 16.8 |
| Negative | 255 | 52.8 |
| Positive (>5%) | 147 | 30.4 |
| Ki67 Status | | |
| Unknown | 56 | 11.6 |
| Negative | 114 | 23.6 |
| Positive (>14%) | 313 | 64.8 |
| Her2 Score | | |
| Unknown | 83 | 17.2 |
| 0 | 119 | 24.6 |
| 1 | 169 | 35.0 |
| 2 | 54 | 11.2 |
| 3 | 58 | 12.0 |

### 2.2.2. Hollywood Movie

Success of a movie was defined as its revenue being greater than 2× the reported budget for the movie, accounting for advertising and promotion budgets which are generally less than the production budget. This dataset came from combining TMDb and MovieLens datasets, accounting for movies included in both sources, gives 3699 movies available for testing and training. The `confidence` attribute for all agents was equally biased between accuracy, precision, and recall. The features found in Table 3 were used in the initial classification comparison testing and were originally selected based on principal component analysis, showing high correlation with predicting movie success. The following features were used for incremental feature addition testing:

- Cast, top 5 listed
- Crew, top 5 listed
- Production company
- Director

### 2.2.3. Airline Satisfaction

We used a publicly available airline passenger satisfaction dataset to test our method on a dataset with a large number of samples, roughly 120,000. Both the breast cancer and Hollywood movie dataset are small, with 483 and 3699 samples, respectively. The airline satisfaction dataset allowed us to verify that our agents are not over-fitting on specific input that may appear frequently and continues to perform in line with other popular classification methods. The `confidence` attribute for all agents was equally biased between accuracy, precision, and recall.

The dataset has 22 input features, seen in Table 4. Passengers reported being either satisfied, neutral, or not satisfied. We considered neutral passengers to be dissatisfied for binary classification purposes. This more evenly split the dataset (48% satisfied and 52% not satisfied) compared with considering neutral passengers satisfied. Features with a possible value of 0 were considered not applicable (N/A). For example, a 0 value for "In-flight WiFi satisfaction" indicates this feature was not present on the airplane, and not an applicable feature for classification for that customer.

**Table 3.** Features available for classification.

| Features | Description |
|---|---|
| budget | given to all agents, reported budget for movie |
| tmdb_popularity | dynamic variable from TMDb API attempting to represent interest in movie |
| revenue | used for sanity checks, reported revenue |
| runtime | unreliable metric for success without including genre information |
| tmdb_vote_average | average score from TMDb, can be combined with ML average |
| tmdb_vote_count | total votes for a movie from TMDb, can be combined with ML count |
| ml_vote_average | average score from ML, can be combined with TMDb average |
| ml_vote_count | total votes for a movie from ML, can be combined with TMDb count |
| ml_tmdb_genres | combined genre information from TMDb and ML; first 2 listed genres used |
| vote_average | combined tmdb_vote_average and ml_vote_average |
| vote_count | combined tmdb_vote_count and ml_vote_count |

**Table 4.** Airline features available for classification.

| Features | Description |
|---|---|
| Gender | Passenger gender: male, female, other |
| Customer type | Loyal or disloyal |
| Age | Customer age |
| Type of travel | Personal or business |
| Seat class | First, business, eco+, eco |
| Flight distance | Distance of journey |
| In-flight WiFi satisfaction | 0 (N/A) 1–5 |
| Flight time convenience | Satisfied with departure/arrival time |
| Ease of online booking | 0 (N/A) 1–5 |
| Gate location satisfaction | 1–5 |
| Food/drink satisfaction | 1–5 |
| Online boarding satisfaction | 0 (N/A) 1–5 |
| Seat comfort | 1–5 |
| In-flight entertainment | 0 (N/A) 1-5 |
| On-board service satisfaction | 0 (N/A) 1–5 |
| Leg room satisfaction | 0 (N/A) 1-5 |
| Baggage handling satisfaction | 0 (N/A) 1–5 |
| Check-in service satisfaction | 1–5 |
| Cleanliness | 1–5 |
| Departure delay | in minutes |
| Arrival delay | in minutes |

### 2.2.4. Sports Betting

The system was tested on sports betting data from January 2021 through November 2022 on college football, National Collegiate Athletic Association (NCAA) Football Bowl Subdivision (FBA), professional football, National Football League (NFL), college

basketball (NCAA Division 1), professional basketball, National Basketball Association (NBA), professional hockey, National Hockey League (NHL), professional baseball, Major League Baseball (MLB), and professional soccer (multiple leagues). The system was tested primarily on "spread" bets, where the favorite needs to win by some pre-defined margin, or the losing team needs to lose by less than some pre-defined margin. It was also tested on "moneyline" bets, where the team that wins the game also wins the bet, with no "spread" taken into account. This is a more common bet type for lower scoring sports like baseball, hockey, and soccer. Moneyline bets are also common in basketball and football when teams are evenly matched and the spread would be very small, less than 2.5–3.5 points. Some testing was done on predicting the total score of a game, but this was not successful for any of the major sports. Professional soccer moneyline bets were tested with three possible outcomes: win, loss, and draw. The moneyline bets for all other tested sports were binary: win or lose. In total, roughly 10,500 bets were tested and the value of each bet was typically around 0.5–1% of 'units' available for betting. For example, if there were 1000 units available, each bet would typically represent 5 to 10 'units'.

Sports betting data were sourced from numerous independent data sources. The data used include historical betting odds data for the NHL, NBA, MLB, NFL, and NCAA basketball and football. Historical odds were also available for the five major European soccer leagues. Historical data for the US MLS soccer league were found to be inaccurate and were not used. Odds data for second-tier European soccer leagues were also found to be inaccurate for certain years and accurate for others. Only years with verified accurate data were included in the training set.

Play-by-play data were available for the NBA, NFL, and MLB. However, these play-by-play data were only used to build season-long statistical data for players and teams when those data were otherwise unavailable through free sources. Some preliminary testing has shown the granularity of play-by-play data increases the noise, which then decreases predictive performance. Roster data, injury information, and game-by-game stats were found on ESPN.com https://www.espn.com/ (accessed on 1 January 2021 through 15 November 2022), Yahoo! Sports https://sports.yahoo.com/ (accessed on 1 January 2021 through 15 November 2022), and The Action Network https://www.actionnetwork.com/ (accessed on 1 January 2021 through 15 November 2022).

ESPN.com was used to determine upcoming games. The Action Network publishes odds from major US-based sports books and was used to determine the best betting odds for upcoming games. For each game, agents were given information about the competing teams. Statistical information for each team was pulled from ESPN or Yahoo! Sports, looking at the past 3, 5, and 10 games, as well as season-long trends and associated stats, e.g., points scored vs. points allowed for NBA teams over the same 3, 5, and 10 games. Roster information for each team was gathered from ESPN and stats for the listed players are populated from both ESPN and Yahoo! Sports. Players season stats, as well as previous 3, 5, and 10 games stats are included. In some sports, match-up specific stats were also available, this included the NBA, NHL, and MLB. MLB stats also included pitcher vs. expected batter lineups from Yahoo! Sports. The current betting line is then pulled from The Action Network and the agents are asked, "will team $a$ win by 5 points" if the spread betting line is "−5". Due to the limitations of making binary predictions, the agents will also be asked the same question on either side of the line until the prediction changes. Both the associated confidence from the swarm step and the distance from the published line when the opinion changes influence the overall confidence in the prediction, with greater distance from the published line giving more confidence in the prediction. For example, if the agents predict team $a$ does not win by 6 points, there is less confidence in the published −5 betting line than when the agents predict team $a$ will win by 9 points.

Predictions from other sources were also gathered and included in the 'meta-swarm' step, as described in Section 2.1.4. Predictions came from ESPN.com (BPI rankings), ESPN.com (fan rankings), Yahoo! Sports (fan rankings), The Action Network ('public' betting, 'expert' picks, 'sharp' action), SportsChatPlace https://scpbetting.com/ (accessed

on 1 January 2021 through 15 November 2022), CollegeFootballNews https://collegefootb allnews.com/ (accessed on 1 January 2021 through 15 November 2022), Winners and Whiners https://winnersandwhiners.com/ (accessed on 1 January 2021 through 15 November 2022) and FiveThirtyEight https://fivethirtyeight.com/sports/ (accessed on 1 January 2021 through 15 November 2022).

## 3. Results

### 3.1. Classification Method Comparison

The following sections show comparisons on three datasets with our WoC-Bots method, the meta-swarm method described in Section 2.1.4 and five popular classification methods: AdaBoost, XGBoost, Random Forest, Deep Neural Network, and Logistic Regression.

#### 3.1.1. Breast Cancer Metastasis

Figure 2 shows the comparison for accuracy, recall, and precision of WoC-Bots with the other classification methods for a breast cancer dataset, predicting node-positive or node-negative disease in breast cancer patients. The results presented here were generated using the clinical features identified in Table 2, but not the additional cellular morphology features described in Section 2.2.1. The leftmost set of columns shows results for the WoC-Bots method, the rightmost set of columns shows results for the "meta-swarm" method described in Section 2.1.4, taking all of the methods listed on Figure 2 as input. The other sets of columns represent popular classification methods. Random Forest produced the best accuracy at 86.9% and the best precision results at 84.0%. The DNN implementation produced the best recall at 92.5%. WoC-Bots outperformed AdaBoost and Logistic Regression in accuracy and recall, but had the second lowest precision results at 72.0% (tied with AdaBoost). The meta-swarm implementation, while not producing the best results for any of the three metrics, improved consistency between the three metrics, produced the second highest accuracy at 84.8%, and the second highest precision at 82.0%.

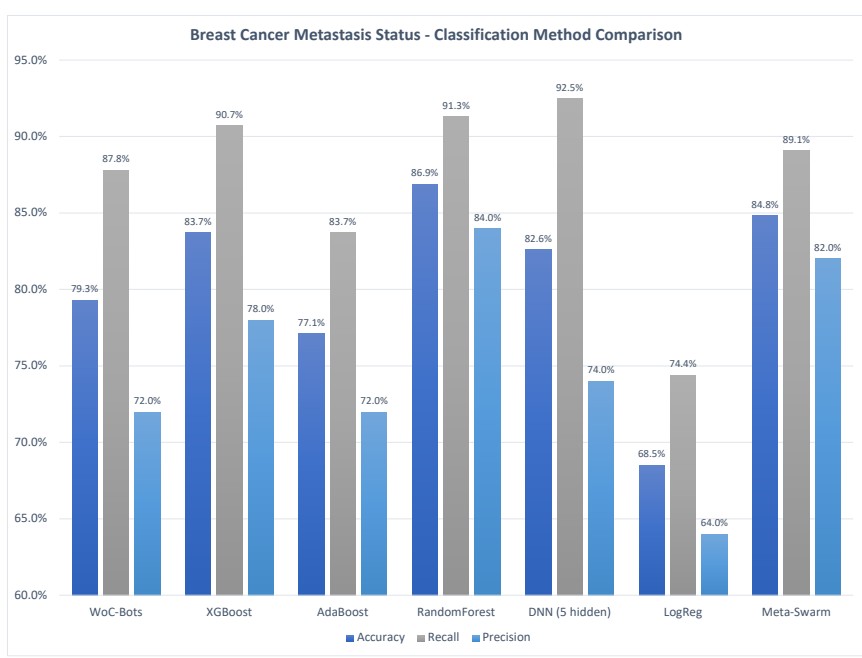

**Figure 2.** Comparison of five classification methods with two versions of the swarm (first and last columns) for the breast cancer dataset.

While WoC-Bots were outperformed by the XGBoost, Random Forest, and DNN methods, WoC-Bots can provide associated confidence values with each prediction. Table 5 shows how many predictions fall into each confidence interval and the intervals associated

accuracy. The combined "Very High", "High", and "Medium" confidence interval captured 78.9% of patients with a combined accuracy of 86.8%. The "High" confidence interval captured 31.9% of patients with an accuracy of 93.1%. WoC-Bots are able to stratify patients into confidence intervals, allowing for improved accuracy within a subset of samples.

**Table 5.** Confidence Interval Distribution and Accuracy—WoC-Bots for Breast Cancer Metastasis Status.

| Interval | $n = 483$ | % of $n$ | Accuracy (%) |
|---|---|---|---|
| Very High Confidence | 3 | 0.62 | 100 |
| High Confidence | 154 | 31.9 | 93.1 |
| Medium Confidence | 224 | 46.4 | 82.3 |
| Low Confidence | 102 | 21.1 | 64.7 |
| Very High + High + Medium | 381 | 78.9 | 86.8 |

Figure 3 presents computation times, in milliseconds, needed to perform a single classification for this dataset using WoC-Bots and other methods. The timing results for all methods were run on the same hardware—AMD Threadripper 1950x CPU, 64 GB of RAM, and 2x Nvidia 2070 GPUs. WoC-Bots and the deep neural network were implemented in Java. The other methods were implemented in Python. As expected, methods which require additional iterative steps take longer to complete, with the WoC-Bots system having the most steps, and taking the most time.

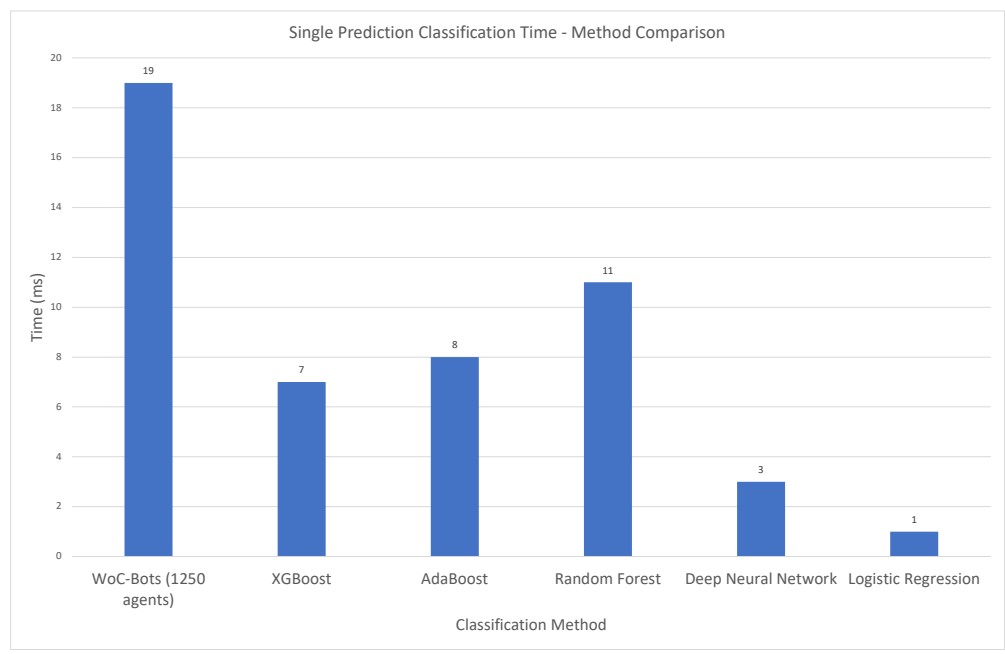

**Figure 3.** Runtime Comparison for a Single Prediction for All Methods.

### 3.1.2. Hollywood Movie Success

Figure 4 presents the same information as Figure 2 for the Hollywood movie success dataset. A movie was considered successful if the revenue was greater than two times the initial budget for the movie. The results presented here use the features listed in Table 3 only, not including the additional cast, crew, production, and director information. WoC-Bots performed very similarly to a DNN implementation, with 81.1% accuracy compared to 81.3% for the DNN. WoC-Bots outperformed the DNN with a precision of 84.0% compared with 82.9% and were outperformed on recall with 79.7% for WoC-Bots and 80.8% for the DNN. The XGBoost and Random Forest methods again outperformed the other classification methods on the accuracy and precision metrics.

The meta-swarm was again tested, outperforming all other methods with an accuracy of 83.7%. Additionally, only the XGBoost method had a better recall value at 82.9% compared with the meta-swarm's recall of 82.6%. Confidence intervals were tested with this dataset, but applied to the output of the meta-swarm instead of directly to WoC-Bots. Table 6 shows the results of this analysis, with 61.4% of samples captured in the "Very High", "High", and "Medium" confidence intervals with a combined accuracy of 87.8%. Additionally, the "Very High" interval was more useful for this dataset, capturing 1.4% of samples with an accuracy of 96.1% and the "High" interval captured over 25% of samples with an accuracy of 91.6%.

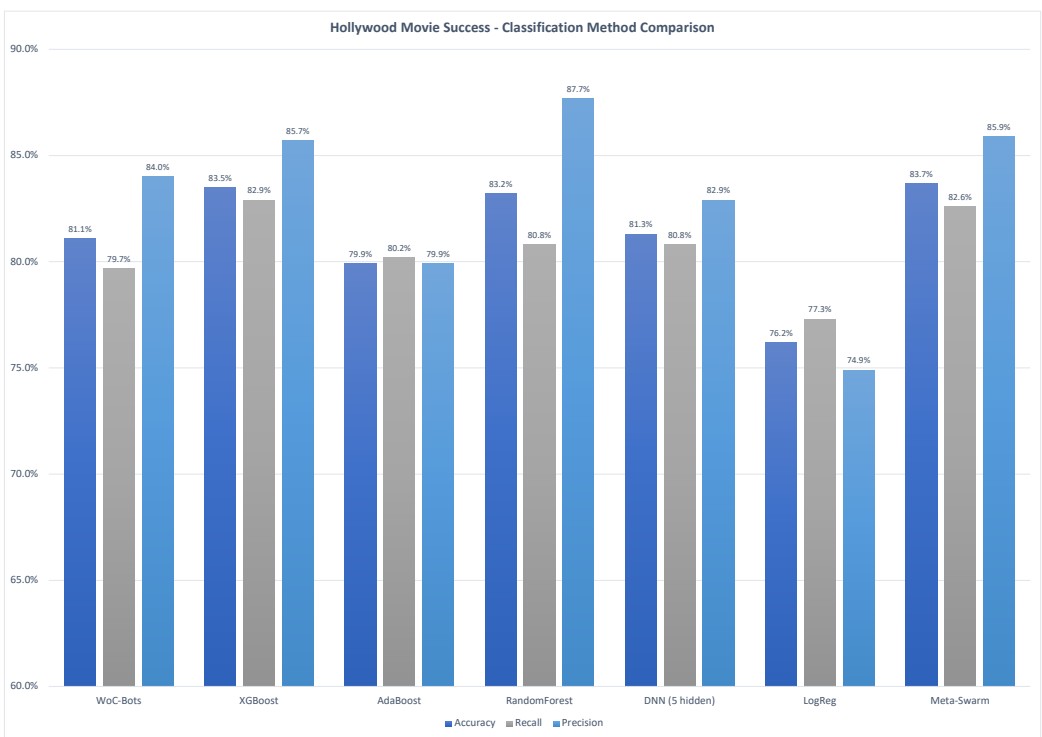

**Figure 4.** Comparison of five classification methods with two versions of the swarm (first and last columns) for the Hollywood movies dataset.

**Table 6.** Confidence Interval Distribution and Accuracy—Meta-Swarm for Hollywood Success.

| Interval | $n = 3699$ | % of $n$ | Accuracy (%) |
|---|---|---|---|
| Very High Confidence | 51 | 1.4 | 96.1 |
| High Confidence | 958 | 25.9 | 91.6 |
| Medium Confidence | 1261 | 34.1 | 84.5 |
| Low Confidence | 1428 | 38.6 | 77.3 |
| Very High + High + Medium | 2271 | 61.4 | 87.8 |

### 3.1.3. Airline Passenger Satisfaction

Figure 5 presents the same classifications methods as Figures 2 and 4. This dataset, however, was far more predictive than the previous datasets, with accuracies over 95% for all methods except logistic regression. Additionally, all methods were consistent with a slighter higher precision than accuracy, and a slightly lower recall than accuracy. WoC-Bots performed similarly to XGBoost, AdaBoost, and Random Forest, better than logistic regression, and slightly worse than the deep neural network. With most methods already performing well, the meta-swarm did not improve over any of them, though, did reduce variability between accuracy, precision, and recall. Table 7 shows the confidence intervals for the meta-swarm for this dataset. Over 15% of samples fell in the "Very High"

interval, over 70% in the "High" interval with the remaining 15% split almost evenly between "Medium" and "Low" confidence. The accuracy values for each interval were very similar, with the "Very High Confidence" interval being outperformed by the "High" and "Medium" intervals. We did not find the confidence intervals to be useful for this dataset; the airline data are highly predictive which made it difficult to stratify samples into confidence intervals.

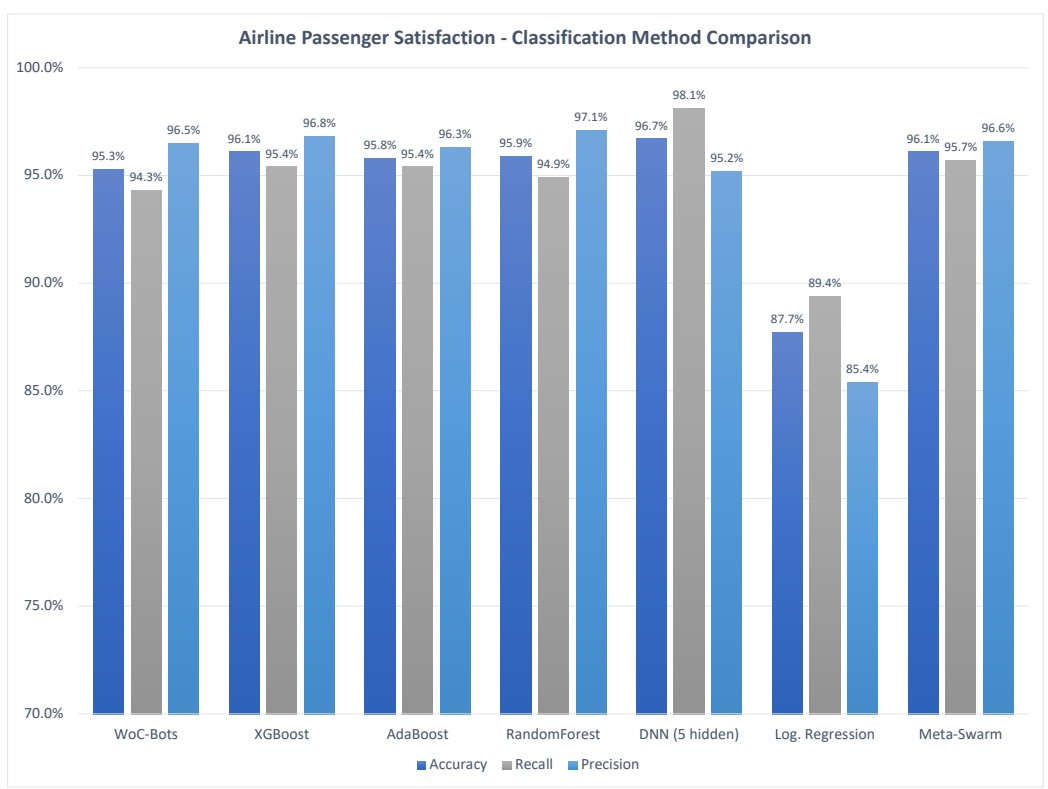

**Figure 5.** Comparison of five classification methods with two versions of the swarm (first and last columns) for the airline passenger satisfaction dataset.

**Table 7.** Confidence Interval Distribution and Accuracy—Meta-Swarm for Airline Passenger Satisfaction.

| Interval | $n = 129,882$ | % of $n$ | Accuracy (%) |
|---|---|---|---|
| Very High Confidence | 19,797 | 15.24 | 95.7 |
| High Confidence | 92,125 | 70.93 | 96.2 |
| Medium Confidence | 9109 | 7.01 | 95.9 |
| Low Confidence | 8851 | 6.81 | 95.7 |

### 3.2. Additional Features

The following sections show the predictive performance improvements produced when including additional features for the breast cancer and Hollywood movies datasets. The goal is to show our method can incorporate new features as they become available without requiring a costly re-training of the full system of agents, and without significant loss of predictive performance compared with a full re-training. This process requires generating new agents, which increases the total number of participating agents. In both datasets, we show results that control for additional predictive performance when including more agents in the simulation.

#### 3.2.1. Breast Cancer Metastasis

Figure 6 shows the original accuracy of our WoC-Bots method for predicting lymph node metastasis status in breast cancer patients. The leftmost column shows the original

results from Figure 2 with 750 agents. The second–from–left column shows the results when using the same clinical features only, but with 1250 agents. The accuracy improved by 0.13%, indicating that increasing the number of agents may minimally improve the accuracy.

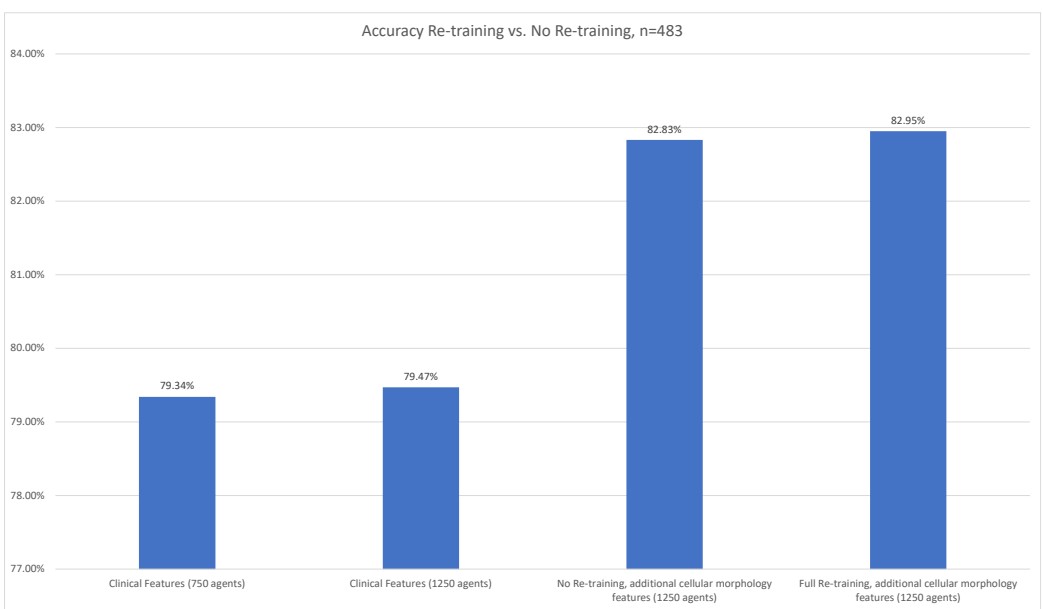

**Figure 6.** Cellular Morphology Features—System Accuracy.

The second-from-right column shows the prediction accuracy after adding the cellular morphology features to the original clinical features. We use the same number of agents (1250) as the previous test and generate additional agents as outlined in Section 2.1.3. In this case, the accuracy increased 3.49% from the original test (left column), and 3.36% as compared with the original clinical features and same number of agents (1250), indicating the accuracy increase largely comes from incorporating the new cellular morphology data rather than from simply generating additional agents. The rightmost column shows the accuracy results after fully re-training the agents with the original clinical features and additional cellular morphology data, with 1250 agents trained on all of the available data. This test shows a small increase in accuracy, 0.13%, compared with the prior method, but requires an expensive re-training of all participating agents. The time and computation cost of re-training depends on the number of agents and available hardware. Using 4x Nvidia GTX 1070 GPUs, an AMD Threadripper 1950x CPU, and an M.2 SSD for storage it takes 2 h, 25 min to generate and train 1250 new agents compared with 35 min to generate and train 500 new agents, a speedup of 4.1×. In testing, available GPU memory will have a large impact on total run time.

### 3.2.2. Hollywood Movie Success

Figure 7 shows similar results when adding new features to the Hollywood movie success test. For this dataset we started with 600 agents and an accuracy of 81.14%. The accuracy did increase more in this test by simply adding more agents, to 81.38%, an increase of 0.25% when moving from 600 to 1000 agents, shown in the second–from–left column. The second–from–right column shows the accuracy after including the additional cast, crew, production, and director information, an increase to 86.77% using the same number of agents, 1000. The rightmost column again shows the accuracy after fully re-training all 1000 agents with all features. We see a slightly larger increase compared with the breast cancer dataset, increasing to 87.81% from 86.77%, an increase of 1.04%.

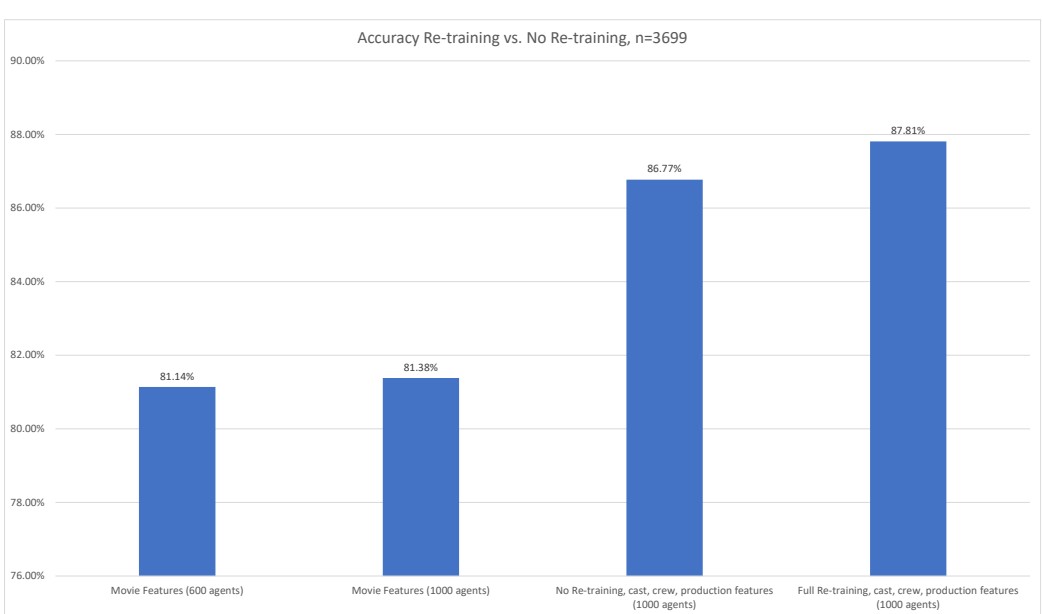

**Figure 7.** Cast, Crew, Production Features—System Accuracy.

It is currently unclear why there was a larger increase when fully re-training agents for this dataset. One possibility is that the additional features were more correlated with the output than in the breast cancer dataset, but this has not been confirmed using principal component analysis. Using the same hardware as described above, it took 12 min to generate and train 400 new agents, and 43 min to fully re-train 1000 agents.

### 3.3. System Distributability

Section 2.1.5 describes the distributed design of the system, with agents allowed to freely move between nodes during the interaction phase and two different methods used during the swarm phase, one that is less compute-efficient, but produces results identical to swarming on a single node, and a second method which is more compute-efficient, but may not produce the same predictive performance as swarming on a single node. WoC-Bots, and the meta-swarm (depending on number of inputs) both can have a large number of participating agents with many decisions directed by a random number generator (which way to move, which node to be distributed to, etc.). Additionally, large multi-agent systems will have variances in their outcomes based on agent-to-agent interactions and agents interacting within their environment [33]. The results presented in this section set a seed to the main random number generator used by the system to guarantee that each run of the system produces the same ordering of interactions and to hold predictive results consistent across multiple runs.

The following tests were conducted using virtual machines with the following hardware:

- CPU: AMD Opteron 6376 (Released November 2012); four virtual CPU (vCPU) cores per virtual machine
- RAM: 12 GB DDR3 per virtual machine
- Ceph-based network storage,

This was with all virtual machines configured with identical virtualized hardware, and with no control over how physical hardware was divided by the virtualization software. For example, when running with four virtualized nodes, those four nodes could be powered by a single hardware CPU, or with each node on a dedicated hardware CPU, or some other possible combination. We found that 12 GB of RAM was enough to allow all agents to maintain themselves in memory without swapping to disk during all phases of this system. Running on more modern, single-node hardware, the interaction step takes an average of 19 ms for 1250 agents and 96 ms for 5000 agents (Threadripper 1950x CPU and 64GB RAM).

### 3.3.1. Runtime Performance

Figure 8 shows interaction run times for 1250 agents on multiple node configurations. Runtime initially increases when using both two and four nodes, with testing showing this is largely due to network transfer penalties associated with frequent writes to the networked storage, as agents traverse nodes. However, nine nodes outperform a single node, with continued runtime improvements when moving to 16 nodes. Figure 9 shows the same node configuration but with 5000 agents participating. There is an increase in overall runtime, with runtime analysis indicating that CPU-bound tasks are the primary cause. The network transfer time takes a similar amount of absolute time in both the 1250 and 5000 agent configurations, indicating there is some initial time penalty. However, we do see a decrease in runtime moving from one node to two nodes, with further decreases in runtime as more nodes are included. Without additional hardware it is difficult to determine how many additional nodes will continue to improve runtime performance.

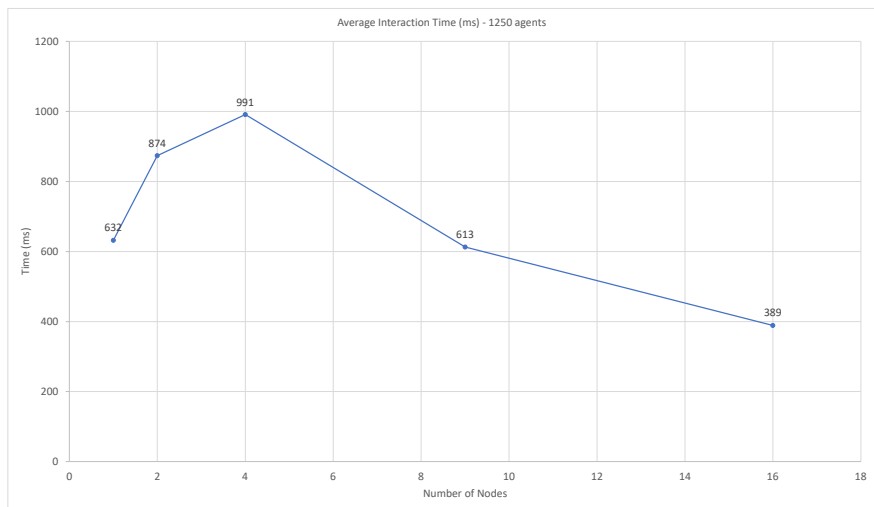

**Figure 8.** Interaction time (ms) for 1250 agents on 1, 4, 9, and 16 nodes, using data from Section 2.2.1 with cellular morphology features.

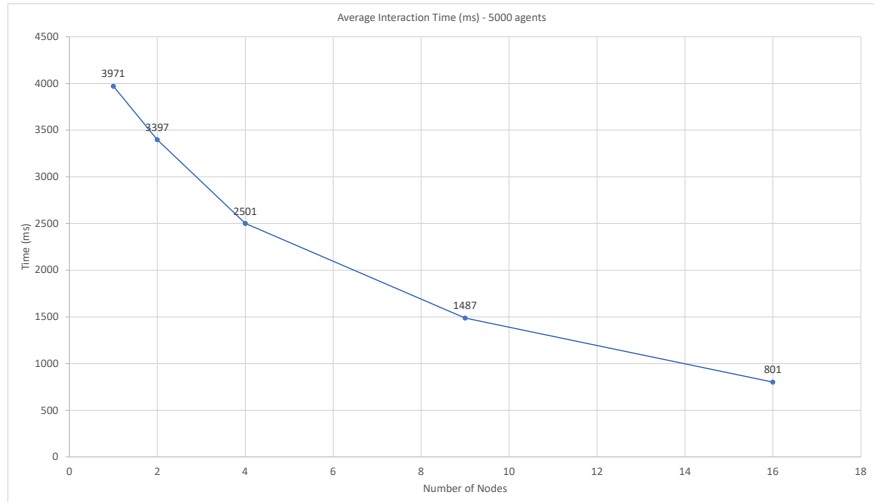

**Figure 9.** Interaction time (ms) for 5000 agents on 1, 4, 9, and 16 nodes, using data from Section 2.2.1 with cellular morphology features.

### 3.3.2. Swarm Performance—Timing and Accuracy

Figure 10 shows the average time the swarm phase takes when allowing agents to move freely between available nodes. Using this method, the swarming phase operates

exactly as it would when a single node is used; however, it is distributed across multiple nodes, with agents frequently moving between nodes. Runtime initially decreases with two nodes, dropping from 9128 ms to 6788 ms, and continues to drop to 5421 ms with four nodes and 4918 ms with nine nodes. However, runtime increases with 16 nodes, to 5301 ms. The swarming phase is not compute-intensive, but when free movement is allowed between nodes it is very network transfer-intensive, which is demonstrated with the increase in runtime when moving from 9 nodes to 16 nodes.

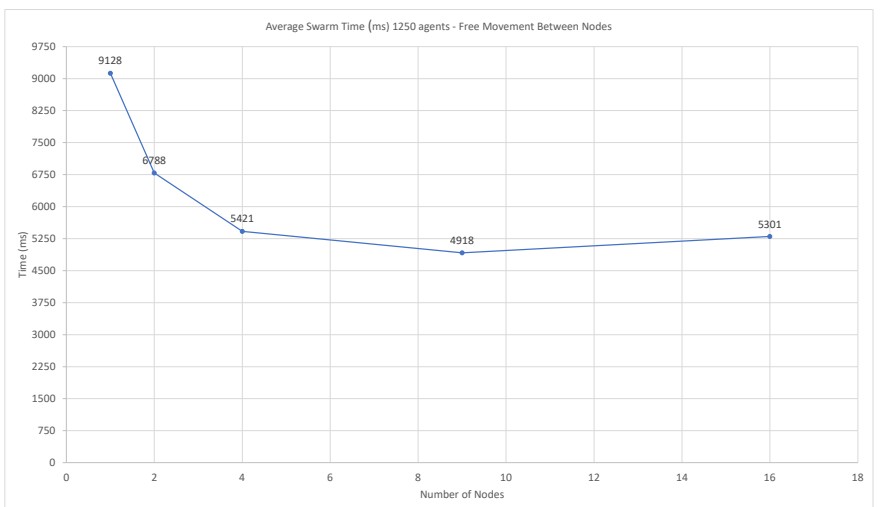

**Figure 10.** Swarm timing (ms) for 1250 agents on 1, 2, 4, 9, and 16 nodes, using data from Section 2.2.1 with cellular morphology features, with agents moving freely between nodes.

Figure 11 shows the average runtime for the swarm phase when disallowing agent movement between nodes. Agents must remain on the node they are initialized on, all swarming processes happen locally on each node, with each node represented in a meta-swarm phase once localized swarming is complete. Using this method we see runtime continue to decrease with additional nodes, indicating the increase in runtime using the previous method was from network transfers between nodes.

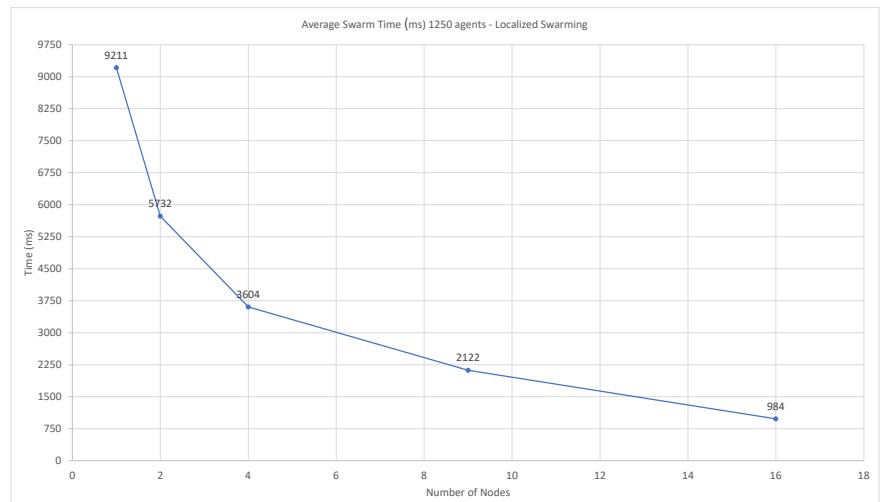

**Figure 11.** Swarm timing (ms) for 1250 agents on 1, 2, 4, 9, and 16 nodes, using data from Section 2.2.1 with cellular morphology features, with node-localized swarming.

Figure 12 shows the prediction accuracy using both of the above methods. In all cases, the predictive performance is decreased when swarming is node-local. The accuracy drops minimally in two of the three examples; dropping from 82.95% to 82.51%, a decrease of

0.44% in the breast cancer dataset and from 95.30% to 94.27%, a decrease of 1.03%, in the airline passenger satisfaction dataset. However, the accuracy drops substantially when predicting Hollywood movie success, going from 81.0% to 74.97%, a decrease of 6.03%. It is currently unclear why this dataset has a larger decrease in predictive performance using the node-local swarming method, a phenomenon that can be studied in future work.

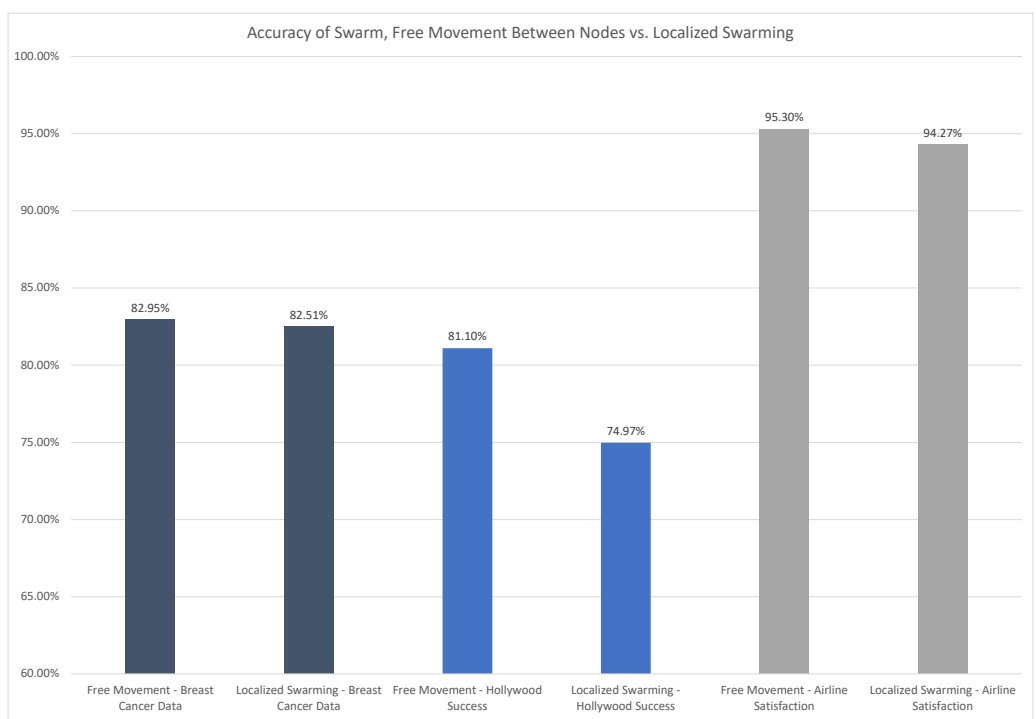

**Figure 12.** Comparison of prediction accuracy for breast cancer, Hollywood success, and airline passenger satisfaction when allowing free movement between nodes vs. node-local swarming.

### 3.4. Sports Betting

Our classification method has been applied to sports betting with some success over a 20-month period and most major sports. The primary focus has been on spread and moneyline bets, with no success in predicting the total score in any major sports. In total, there have been 10,500+ bets tested with the value of each bet being between 0.5% and 1.0% of available units. The system started with 150 units and has accumulated over 11,300 units in the past 20 months. As expected in a highly volatile, real-world use case, there is variability in the results over shorter periods of time, but over any 60+ day period of time the results are strictly increasing.

On 1 October 2021, the system switched to the 'meta-swarm' mode instead of relying only on predictions made by WoC-Bots. Other sources of information included in the calculation are described in Section 2.2.4. At all times the predictions from the system were allowed to be ignored in extraordinary cases, such as late-breaking injury news or unexpected local events, e.g., city-wide protests in some event location. Less than 3% of predictions from the system were ignored. When predictions were ignored, no action was taken on the event. Other news and information was assumed to be accounted for in the betting line offered, injury reports, and team roster information for the event.

Figure 13 shows the total units returned by the system over this time period. Note that around 1 June 2021 the system did briefly go negative. Over this period of the year the only major sports being played are MLB and soccer, both sports that have proven to be more volatile and more difficult for this system to predict. Additional volatility was suspected to come from data inconsistencies. All major sports in the preceding year disallowed fans, had shortened seasons, or were playing condensed seasons to account for lost time due to the COVID-19 pandemic. This impacted sports and teams differently, with a larger impact

seen in cardio-intensive sports such as soccer, and on teams with older players as compared with the league average. In all sports, except the NHL, the home field advantage enjoyed prior to the COVID-19 pandemic was reduced.

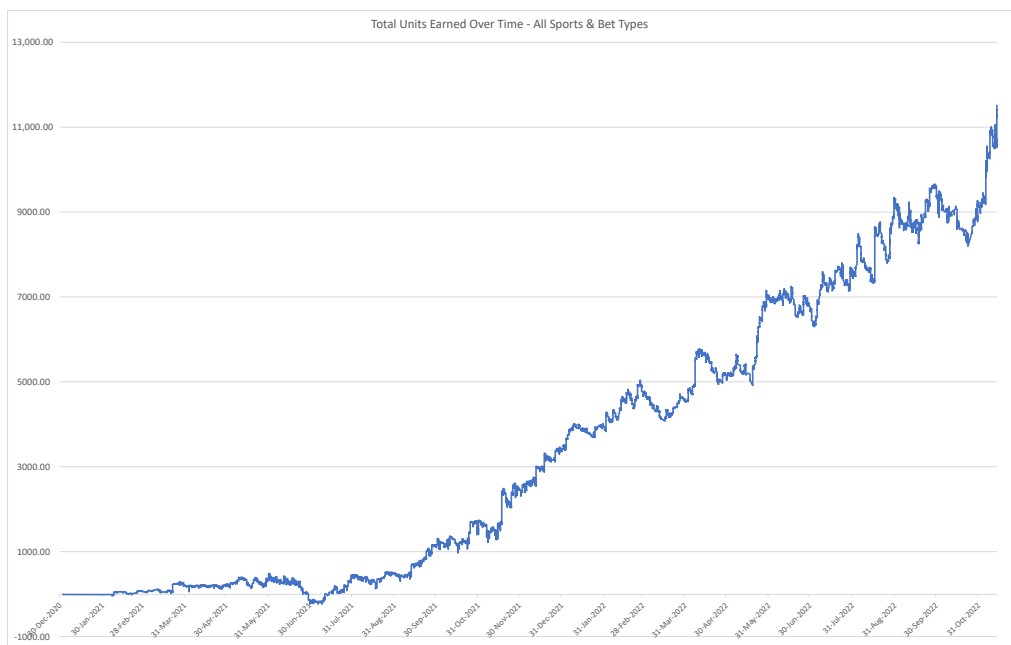

**Figure 13.** Total units earned over time for all sports tested on, moneyline and spread bets, 1 January 2021 through 15 November 2022.

Tables 8 and 9 show the total units risked, units returned beyond the units risked, the return on investment (ROI), the total bets, total winning bets, and the win rate for WoC-Bots (1 January 2021 through 30 September 2021) and the meta-swarm (1 October 2021 through 15 November 2022). In sports betting, the ROI is dictated not just by a win or loss, but also by the odds offered by the sports book. Both WoC-Bots and the meta-swarm have tended to prefer underdog bets with better odds. Bets on underdogs have positive odds; this means if the bet wins you receive more units back than were risked. For example, if 10 units are risked on odds of +220 and the bet wins you will receive 32 units, the original 10 risked units, and 22 additional units for the win. Spread bets are typically set with odds at −110, meaning if 10 units are risked and the bet wins you receive 19.09 units back, the original 10 risked units and 9.09 additional units for the win.

**Table 8.** WoC Bets: Risk, Return and Win Rate (Accuracy) for Spread and Moneyline Bets.

| Values | Spread Bets | Moneyline Bets |
| --- | --- | --- |
| Units Risked | 8212 | 18,532 |
| Units Returned | 628.5 | 668 |
| ROI | 9.5% | 3.6% |
| Total Bets | 1161 | 3282 |
| Winning Bets | 572 | 1481 |
| Win Rate | 49.06% | 45.12% |

The win rate in both tables can be thought of as the accuracy of the predictions. Notice that the win rate is below 50% for spread and moneyline bets using both WoC-Bots and the meta-swarm. The default spread line offered by a sports book will be set to −110 odds, but most sports books allow betters to pick their own spread line with different odds. For example, a default line could be `team-xyz` (−5.5) for −110 odds, meaning that `team-xyz` needs to win by six points to win the bet. The sports books will also offer options such as (−6.5) for +105 odds or (−7.5) for +115 odds. When the prediction has a high associated

confidence value we can take a custom line with positive odds on a spread bet, allowing for a win rate below 50% to still be profitable. This can also be seen for moneyline bets; the win rate for moneyline bets using the meta-swarm actually drops, from 45.15% using WoC-Bots only, to 40.19%. However, the ROI increases from 3.6% using WoC-Bots to 6.7% using the meta-swarm. The system became better at selecting underdog winners, increasing the ROI despite the decrease in win rate.

**Table 9.** Meta-Swarm: Risk, Return and Win Rate (Accuracy) for Spread and Moneyline Bets.

| Values | Spread Bets | Moneyline Bets |
|---|---|---|
| Units Risked | 48,159 | 80,408 |
| Units Returned | 4863.5 | 5364.5 |
| ROI | 10.1% | 6.7% |
| Total Bets | 2376 | 3225 |
| Winning Bets | 1174 | 1296 |
| Win Rate | 49.33% | 40.19% |

Some testing has been done to improve win rate by biasing the agents to prefer favorites. This testing has shown poor results, with negative ROI despite an increase in win rate. The only exception to this is the last 15–20% of both the NBA and MLB seasons, where many teams know their playoff positions and the favorites tend to win more frequently than during the early and middle parts of the seasons. This is, however, reflected with worse odds on favorites, which requires a significantly higher win rate to maintain a positive ROI.

## 4. Discussion

Our classification system has demonstrated competitive performance with other state-of-the-art classification methods for multiple different datasets with a varied number of samples and input features. While Random Forest and a Deep Neural Network implementation outperformed WoC-Bots in maximum accuracy on the tested datasets, WoC-Bots outperformed AdaBoost and Logistic Regression methods in two of the three tested datasets and performed similarly to all of the top methods in the third dataset. WoC-Bots' accuracy, precision, and recall metrics tracked similarly with the other classification methods; XG-Boost, AdaBoost, Random Forest, DNN, and WoC-Bots all showed higher recall scores and lower precision scores on the breast cancer dataset. Similarly, the same methods showed higher precision and lower recall scores for the Hollywood dataset, indicating WoC-Bots are learning similar information as the other classification methods. While WoC-Bots prediction runtime is the slowest of the methods shown in this paper, runtime optimization has not yet been a focus. This is something that can and should be improved in future work; for example, the interaction arena interactions can be optimized to a series of matrix multiplication operations for each pair of interacting agents. Further, this optimization step would also reduce the runtime during the swarming phase. The current system allows for easy behavioral modification while testing a new method, but this is an intended future optimization.

The multi-agent design allows for new, previously unknown features to be included in any existing classification problem without re-training the full network (or set of existing agents). We demonstrated significant accuracy improvements when adding new features, while also demonstrating similar overall performance when compared with re-training the agents with all of the available features. This method is useful when the final set of features is unknown or constantly expanding. The real-world results we have shown with sports betting made use of this feature frequently to incorporate new data from recent games, as new data sources were discovered or new metrics were developed to track player and team performance. Individual WoC-Bots were quickly updated to include results from the most recent games and new bots were generated to represent novel data sources or statistical metrics.

WoC-Bots can be distributed across multiple hardware nodes, with no requirement for a single powerful machine to train a very deep and wide monolithic neural network. We demonstrated improved runtime performance when adding compute nodes for both the interaction and swarming steps of the system. The 'meta-swarm' system developed to represent the prediction of each of the nodes has greatly reduced the runtime of the distributed swarm step by allowing each agent to stay on a single node throughout the process. This system has shown similar accuracy to a single-node swarm in two of the three tested datasets.

A by-product of developing the distributed 'meta-swarm' was the ability to include additional prediction sources into the overall prediction, after the WoC-Bots have made their initial prediction. We have shown this reduces the overall variance in accuracy, precision, and recall in all of the datasets tested, while also producing the best accuracy of all methods tested on the Hollywood dataset. This system, primarily using the 'meta-swarm' extension, has been applied to a real-world problem, making predictions about the outcomes of sporting events.

## 5. Conclusions

We have demonstrated a multi-agent, binary classification system based on Wisdom-of-Crowds which uses a honey-bee-derived swarm mechanism for opinion aggregation. Our system is competitive with other, state-of-the-art, classification methods when tested on three different datasets. The multi-agent design allows for multiple, significant, advantages over the other classification methods; WoC-Bots can be distributed across multiple hardware nodes, incrementally include new features without re-training existing agents, and the aggregation mechanism is flexible enough to incorporate predictions from other sources. Further, the aggregation mechanism can provide confidence values for each prediction, which allow us to stratify samples into confidence categories, significantly improving the average accuracy, precision, and recall for a subset of samples in two of the three tested datasets. The meta-swarm, where we incorporate predictions from other classification methods and sources, improved the variance between the three metrics (accuracy, recall, precision), and had the best overall accuracy when predicting the success of Hollywood movies.

We applied our method to a real-world sports betting problem, producing consistent return on investment using both WoC-Bots directly as well as the meta-swarm, incorporating the predictions from WoC-Bots and other freely available prediction sources. Over a period of nine months, we saw an ROI of 9.5% on spread bets and 3.6% on moneyline bets using WoC-Bots alone, and an ROI of 10.1% on spread bets and 6.7% on moneyline bets over a year of using the meta-swarm.

Future work should focus on improving the performance when distributing the interaction and swarm steps by optimizing the network transfers. Additional investigation should also focus on why the meta-swarm step did not perform well on the Hollywood dataset, aimed at determining why the distributed performance was significantly worse than single-node swarming when a similar performance drop was not present in the breast cancer or airline passenger satisfaction datasets. It is unclear if this is a fundamental issue with some types of data or if this was a limitation of a specific dataset.

**Author Contributions:** Conceptualization, S.G.; methodology, S.G. and D.E.B.; software, S.G.; investigation, S.G. and D.E.B.; resources, S.G. and D.E.B.; data curation, S.G.; writing—original draft preparation, S.G.; writing—review and editing, S.G. and D.E.B.; supervision, D.E.B. All authors have read and agreed to the published version of the manuscript.

**Funding:** This research received no external funding.

**Institutional Review Board Statement:** The study to gather the breast cancer dataset was conducted in accordance with the Declaration of Helsinki, and approved by the Institutional Review Board of Drexel University College of Medicine for studies involving humans. Protocol code: 1411003203A001. Approval date: 2014-11-03. Investigator: David Edward Breen. Title: Online Digital Storage Array

for Archived Histology Slide Imaging. IRB approval is not applicable for the Hollywood, airline passenger satisfaction, or sports betting data.

**Informed Consent Statement:** Informed consent was obtained from all subjects involved in the study to gather the breast cancer dataset.

**Data Availability Statement:** Some, but not all, of the data used in this publication are made available. The breast cancer dataset is being withheld due to patient privacy concerns. The sports betting data, while available from public sources, are also being withheld due to discrepancies in various jurisdictions on the legality of sports betting. The transformed dataset used for Hollywood success prediction can be found at https://data.mendeley.com/datasets/gj66mt4s4j/2 (accessed on 21 October 2019) and the dataset used for airline passenger satisfaction can be found at https://data.mendeley.com/datasets/8ppmphw235 (accessed on 20 December 2022). The code will be made available upon reasonable request Sean Grimes, spg63@drexel.edu.

**Acknowledgments:** The authors would like to thank Mark D. Zarella, and Fernando U. Garcia, for their support and guidance in refining this approach for use with breast cancer patients and for help in understanding how to best use the available patient features. The authors would also like to thank ESPN, Yahoo Sports!, The Action Network, SportsChatPlace, CollegeFootballNews, Winners and Whiners, and FiveThirtyEight for making sports data available and their analysis of the sporting world.

**Conflicts of Interest:** The authors declare no conflict of interest.

## Abbreviations

The following abbreviations are used in this manuscript:

| | |
|---|---|
| MDPI | Multidisciplinary Digital Publishing Institute |
| DOAJ | Directory of open access journals |
| WoC | Wisdom of Crowd |
| MLP | Multi-layer Perceptron |
| ANN | Artificial Neural Network |
| DNN | Deep Neural Network |
| CNN | Convolutional Neural Network |
| RNN | Recurrent Neural Network |
| PM | Prediction Market |
| API | Application Programming Interface |
| DL4J | DeepLearning4j |
| JVM | Java Virtual Machine |
| UWM | Unweighted Mean Model |
| WVM | Weighted Voter Model |
| TMDb | The Movie Database |
| NCAA | National Collegiate Athletic Association |
| FBS | Football Bowl Subdivision |
| NFL | National Football League |
| NBA | National Basketball Association |
| NHL | National Hockey League |
| MLB | Major League Baseball |
| MLS | Major League Soccer |
| ROI | Return on Investment |

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
