# Peer review of "A Multi-Agent Approach to Binary Classification Using Swarm Intelligence"

_futureinternet, doi:10.3390/fi15010036_

Round 1
Reviewer 1 Report
In this paper, the authors present a method for binary classification implemented as a multi-agent system. The system can be distributed across multiple compute nodes, using agents each with partial knowledge of the information space. The presented system is based on Wisdom of Crowds and uses a honey-bee-derived swarm mechanism for opinion aggregation, that provides a prediction and associated confidence value. In this way, the multi-agent approach allows new features to be incorporated into classification problems without re-training the existing agents, reducing the computational costs and training time of adding new feature to an existing model. The authors show how this method compares with other common classification methods using three datasets, predicting the metastasis status of breast cancer patients, the success of Hollywood movies, and the satisfaction of airline passengers.
The problem discussed in this article corresponds perfectly to the topic of the Journal "Future Internet". The idea of the authors to tackle this problem is very interesting and deserves attention. Methodologically, the paper is well structured and logically following the approach chosen by the authors.
The authors' idea to propose a binary classification method implemented, as a multi-agent system, where each agent represents an autonomous perceptron, is very interesting and deserves attention. The point of conducting such a study is very well motivated. Convincing evidence for the usability and effectiveness of the proposed method is presented - tests with three different datasets have been conducted, and the results are discussed extensively and in detail in this paper. Furthermore, a comparative analysis with existing well-known classification algorithms is done. The advantages of such an agent-based solution are discussed in detail.
Perhaps it will be useful for interested readers if the authors provide some information about the development tools with which the system was implemented - I did not find such information in the text of the article.
Reviewer 2 Report
The article proposes a Wisdom-of-Crowds-Bots based binary classification approach. The main advantage of the proposed principle is that each agent contains a rather simple neural network - a multi-layer perceptron with relatively low number of hidden layers and processes part of the features. The final classification is obtained by combining the opinions from large number of agents forming a swarm. Classical Swarm approaches are used for doing that. The results of the prediction in terms of accuracy, precision and recall are comparable to the existing classification approaches. From the perspective of the classification results it is yet another prediction model. Still, it has an important advantage - it is possible to add new features to the model without retraining the whole model and also to include new external knowledge from other sources which is very important in case there is a need to adapt the model for situation when other features are available by retraining only part of the model (in particular - newly added agents). It seems that one of the disadvantages of the proposed approach is the complex classification process that takes time/resources. The authors of the paper discuss how this process can be distributed over multiple computational nodes to speed it up, still the paper would benefit from comparing the time necessary to do one prediction by the proposed model and the ones used for comparison just to demonstrate how big the difference is. Otherwise the method has been extensively tested on four different predictions - three different datasets (compared with other methods) as well as one real world scenario of sports betting demonstrating a decent ROI in two years period.
Minor additional remarks:
*) Choice of other machine learning models for comparison should be justified somehow. The choice looks reasonable for the reviewer, but justification is missing.
*) What is the most important metric for evaluating the learning results for each dataset? The reviewer believes that the article would benefit from discussion for which datasets accuracy is most important, for which precison is the main metrics to consider and maybe for some of them recall is the most important.
*) Several values are chosen voluntarily, for example the percentage of agents agreeing on the same prediction for each confidence level. Such decisions should be justified in the article.
*) The introduction section should end with an outline of the paper showing what each section of the paper contains. Consider adding this to the existing paragraph starting at line 59.
*) Section 2.1.1. should not start with a table.
*) Line 753: NBA - National Basketball Association, not League
Overall the paper is well written and the quality of English is good.
